# Robotics in Physical Rehabilitation: Systematic Review

**DOI:** 10.3390/healthcare12171720

**Published:** 2024-08-29

**Authors:** Adriana Daniela Banyai, Cornel Brișan

**Affiliations:** Department of Mechatronics and Machine Dynamics, Technical University of Cluj-Napoca, 400114 Cluj-Napoca, Romania; adriana.tomsa@campus.utcluj.ro

**Keywords:** motor rehabilitation, robotic care, robot-assisted therapy

## Abstract

As the global prevalence of motor disabilities continues to rise, there is a pressing need for advanced solutions in physical rehabilitation. This systematic review examines the progress and challenges of implementing robotic technologies in the motor rehabilitation of patients with physical disabilities. The integration of robotic technologies such as exoskeletons, assistive training devices, and brain–computer interface systems holds significant promise for enhancing functional recovery and patient autonomy. The review synthesizes findings from the most important studies, focusing on the clinical effectiveness of robotic interventions in comparison to traditional rehabilitation methods. The analysis reveals that robotic therapies can significantly improve motor function, strength, co-ordination, and dexterity. Robotic systems also support neuroplasticity, enabling patients to relearn lost motor skills through precise, controlled, and repetitive exercises. However, the adoption of these technologies is hindered by high costs, the need for specialized training, and limited accessibility. Key insights from the review highlight the necessity of personalizing robotic therapies to meet individual patient needs, alongside addressing technical, economic, social, and cultural barriers. The review also underscores the importance of continued research to optimize these technologies and develop effective implementation strategies. By overcoming these challenges, robotic technologies can revolutionize motor rehabilitation, improving quality of life and social integration for individuals with motor disabilities.

## 1. Introduction

According to the World Health Organization, approximately 15% of the global population, over 1 billion people, live with some form of disability. Between 110 and 190 million of these individuals experience significant difficulties in functioning [1]. These disabilities can result from strokes, traumatic craniocerebral injuries, neurological conditions, and degenerative diseases, severely impairing people’s ability to perform essential daily activities such as walking, bathing, eating, and dressing, thus reducing their independence and quality of life. Beyond physical impacts, motor disabilities often also contribute to the development of significant psychological issues, including increased rates of anxiety and depression, highlighting the acute need for comprehensive rehabilitation approaches that address both the physical and psychological needs of patients [2].

The increasing prevalence of motor disabilities, accompanied by an ageing population in many countries, is placing growing pressure on healthcare and social assistance systems [1,2]. Motor rehabilitation aims to restore or improve the motor functions of patients through specific exercises, physical therapy, and advanced technology such as robotic systems. Robotic systems can significantly improve strength, coordination, and dexterity, enabling patients to perform daily activities independently, which is essential for self-esteem and social integration.

Motor rehabilitation also stimulates neuroplasticity, the brain’s ability to reorganize and form new neural connections in response to learning and experience. Repeated and controlled therapies can facilitate the relearning of lost motor skills and contribute to functional recovery after injuries to the nervous system. 

By maintaining and improving mobility, rehabilitation contributes to overall health maintenance and prevents further deterioration of patients’ conditions.

Advances in robotics and artificial intelligence have opened new perspectives for motor rehabilitation, offering hopes for more effective and personalized treatments. Robotic systems, from exoskeletons to end-effector-based assisted training devices, introduce a level of precision, repeatability, and adaptability unprecedented in traditional therapy and promise significant improvements, while also providing sensory and motor feedback. They allow for controlled and personalized therapeutic exercises tailored to the progress and needs of each patient, contributing to more efficient individual recovery. Robotic systems are also key in providing visual, auditory, and haptic feedback to patients. Haptic feedback integrates the sense of touch into digital experiences by employing actuators that generate tactile sensations through vibrations, forces, and motions, simulating real-world interactions. This technology utilizes precise control algorithms and advanced materials to deliver accurate and responsive tactile stimuli, enhancing user interface interactions during therapeutic exercises, supporting robotic systems, and amplifying the process of neuroplasticity [3]. 

Robotic technologies enable precise and real-time monitoring of patient performance and progress. These data can be used to adjust and optimize therapy, ensuring that patients receive the optimal level of assistance and challenge throughout rehabilitation sessions.

Robotic rehabilitation has been transformed into a more engaging and motivating experience for patients by integrating elements of gaming and virtual reality into exercises. This playful approach can improve treatment adherence and contribute to the emotional and psychological well-being of patients.

Implementing robotic systems in the motor rehabilitation of and care for patients with physical disabilities can reduce the workload of therapists, allowing them to focus on aspects that require direct and personalized human intervention, thus improving the overall efficiency of the rehabilitation and care process. Despite these advantages, challenges remain, such as high equipment costs and the need for specialized staff training and adaptations to clinical infrastructure. The significant potential for improving patient outcomes justifies investments in research and development in this field. As a result, initiatives such as the DDSKILLS (Digital and Social Skills for All) project have been established, which exemplifies a commitment to harnessing emerging technologies to support both professional caregivers and individuals with disabilities. DDSKILLS has developed training courses focused on improving the digital skills of carers to make more effective use of new technologies and to support people with disabilities in their use [4].

The integration of emerging technologies, including wearable technology and the Internet of Things (IoT), in motor rehabilitation promises to enhance the quality of therapeutic interventions and improve the lives of patients with physical disabilities.

Rigorous evaluation and effective implementation strategies are essential to ensure that these innovations benefit a board spectrum of patients. These strategies emphasize the opportunities that robotic technology offers to shape the future of motor rehabilitation, enhancing the quality of therapeutic interventions and contributing to a future in which every patient has the chance for an improved life [5].

The overall objective of this systematic review is to conduct an in-depth review of the manners in which innovations in robotics rehabilitation systems can facilitate motor recovery and reduce the workload of caregiving staff, thereby improving the quality of life for patients with physical disabilities. The specific objectives are as follows:Exploring the diversity of robotic technologies applied in motor recovery, from exoskeletons and robotic arms to personal assistant robots and brain–computer interfaces;Analyzing the advantages and limitations of each category of robotic systems in the specific context of motor recovery;Investigating the technical, economic, social, and cultural barriers that may limit access to or acceptance of robotic technology for motor recovery;Identifying facilitating factors that promote the adoption of robotic technologies, including funding initiatives, training programs for medical staff, and awareness campaigns.

By achieving these objectives, this study provides a solid foundation for understanding the complexity and potential of robotic technologies in improving the motor recovery process and supporting patients with motor disabilities. It contributes to the development of effective implementation strategies and the optimization of care practice.

Robotics has emerged as a transformative technology across various healthcare domains, extending far beyond its initial applications in surgical assistance. For instance, robotic systems like the Da Vinci robot have revolutionized minimally invasive surgery by providing surgeons with enhanced precision and control. In patient care, assistive robots are increasingly used to support daily activities for the elderly or individuals with disabilities, such as helping with mobility, feeding, and dressing. Additionally, in the field of telemedicine, robots have played a crucial role in remote monitoring and consultations, especially in underserved regions and during global health crises like the COVID-19 pandemic. The diversity of these systems, ranging from surgical robots to assistive and telemedicine robots, highlights their potential to address a variety of healthcare challenges. While this review will focus primarily on the application of robotic systems in motor rehabilitation, it is essential to recognize that these technologies are part of a much larger movement towards the integration of robotics in healthcare, promising to enhance both the quality and accessibility of medical care across the globe.

## 2. Materials and Methods

This review was performed in accordance with the PRISMA (Preferred Reporting Items for Systematic Reviews and Meta-Analyses) guidelines (Figure 1). 

### 2.1. Literature Search

A comprehensive literature search of several databases, including PubMed, IEE and Web of Science, was conducted to identify relevant studies on the use of robotic technologies in motor rehabilitation of physically disabled people. The search strategy utilized a combination of keywords such as “robotic technology”, “motor rehabilitation”, “physical disabilities”, and “exoskeletons”.

Thus, a total of more than 100 articles dealing with this topic were found in the literature. The present article does not list all the articles found, since after applying the selection criteria, the percentage of articles selected for analysis was less than 50%, and they are listed in the list of bibliographical references. 

### 2.2. Inclusion and Exclusion Criteria

Inclusion criteria: studies published within the last 15 years, focused on the use of robotic systems in motor recovery or patient care for people with physical disabilities, reviewing and analyzing published solutions with empirical data;Exclusion criteria: articles without empirical data, studies that did not specifically address the use of robotic technology in motor recovery or care for physically disabled patients, and studies conducted on animals or that did not involve human subjects.

### 2.3. Data Extraction

Data extraction involved the systematic retrieval of detailed information on study design, sample size, participant demographics, type of robotic technology used, intervention protocols, rehabilitation outcomes, follow-up duration, and any reported challenges or limitations. These data were organized and synthesized to address the established objectives.

### 2.4. Quality Assessment

The quality of the included studies was assessed using standardized criteria (Preferred Reporting Items for Systematic Reviews and Meta-Analyses). These guidelines are used to improve the reporting of systematic reviews and meta-analyses to ensure the reliability and validity of the findings. They include a 27-item checklist and a four-phase flow diagram. Jadad Scale is a procedure to independently assess the methodological quality of a clinical trial, particularly in the field of pain management. It focuses on randomization, blinding, and accounting of all patients.

Specific criteria were applied to evaluate the risk of bias in different domains, including selection bias, performance bias, detection bias, and reporting bias. Each study was assessed accordingly and discrepancies between assessors were resolved through discussion and consensus.

### 2.5. Risk of Bias

The risk of bias assessment was conducted to ensure the quality and reliability of the studies included in the review. Standardized tools were used to analyze potential sources of bias at different stages of the studies, including participant selection, allocation to interventions, outcome measurement, and data reporting. Each study was independently assessed by two reviewers and any discrepancies were resolved through discussion and consensus. This rigorous assessment allowed us to identify and control for variables that might influence the results and conclusions of the review, thus ensuring the best possible picture of the effectiveness and applicability of robotic technologies in motor rehabilitation.

### 2.6. Data Synthesis

The extracted data were synthesized to provide a comprehensive overview of the current state of the field of robotic technologies in motor rehabilitation. Both qualitative and quantitative analyses were performed, summarizing the main findings, identifying trends, and highlighting research gaps. This synthesis aimed to present a coherent and informative summary of the evidence base.

The selection process was carried out in stages, initially screening titles and abstracts to identify potentially relevant studies. Subsequently, the full texts of these studies were thoroughly evaluated, applying inclusion and exclusion criteria, to determine their final eligibility for analysis.

In Table 1, the selected articles for review based on the inclusion criteria are presented. These articles include systematic reviews, meta-analyses, randomized controlled trials, case and pilot studies, exploratory studies, and reviews of technical developments and robotic systems for patient care.

The extracted information was organized and synthesized to address the objectives set. Qualitative or quantitative analyses, as appropriate, were conducted to assess the effectiveness and impact of robotic interventions on motor recovery, while identifying trends, challenges, and future research directions in this area.

## 3. Classification of Robotic Systems for the Rehabilitation and Care of Patients with Disabilities

According to the current state in medical exercise and rehabilitation, robotic systems are defined as devices that integrate advanced motors and control systems to assist or perform therapeutic movements. These systems often feature real-time feedback and adaptive control mechanisms to tailor the exercises to the patient’s needs. These robots provide interactive feedback, adjust difficulty levels based on patient performance, and aim to enhance motivation and adherence to rehabilitation programs [50,51]. 

Exoskeletons and wearable robotics are wearable devices equipped with actuators, sensors, and control systems designed to enhance or restore the movement capabilities of individuals with mobility impairments. These devices are often used in rehabilitation to provide support, improve motor function, and facilitate physical exercise [52].

The detailed classification provides a comprehensive coverage of robotic systems for motor rehabilitation, taking as criteria the objectives formulated in the previous paragraph: exploration of technologies, analysis of advantages and limitations, investigation of barriers, and identification of facilitating factors.

### 3.1. The Diversity of Robotic Technologies Applied in Motor Rehabilitation, Including Exoskeletons, Robotic Arms, Personal Assistant Robots, and Brain–Computer Interfaces 

Type of assistance:
In upper limb motor rehabilitation, devices that are designed to assist and improve the motor function of the arms and hands are used, providing repetitive and controlled exercises that aid in recovery after strokes or other conditions. Examples include InMotion robotic arms, the ARMin exoskeleton, and robotic gloves such as HandSOME.In lower limb motor rehabilitation, robotic lower limb systems facilitate gait rehabilitation by supporting the body weight and guiding leg movements in a way that mimics natural walking. Examples include Lokomat and the ReWalk exoskeleton.Some robotic systems are designed to help people with reduced mobility to perform various daily activities, such as feeding, dressing, and handling objects. Examples include the PR2 personal assistance robot and the JACO robotic system.Some interactive robots, such as NAO and Pepper, and the PARO robot are used to improve social interaction and communication skills, especially among individuals with autism or neurodegenerative conditions. These robots are intended to enhance social and emotional skills, provide cognitive stimulation, improve well-being, and reduce feelings of isolation.Type of rehabilitation therapy:
In passive rehabilitation therapy, the robot guides the patient’s limb through a range of motions without the patient exerting any effort. This approach is essential in the early stages of rehabilitation when the patient has limited mobility or reduced muscle strength.In active rehabilitation therapy, the patient initiates the movement, and the robot helps only when necessary. This method stimulates neuroplasticity and encourages the relearning of motor skills through active practice.Bilateral rehabilitation therapy involves the simultaneous use of both limbs, where the activity of a healthy limb is mirrored or assisted by the robot on the affected limb. This type of therapy is used to improve coordination and movement symmetry between the upper or lower limbs.User interaction mode:
Direct interfaces allow users to interact directly with the device through physical buttons, touchpads, or gesture control. This includes rehabilitation systems that use touch screens or motion sensors to capture and respond to user actions, such as the HapticMaster Robotic Arm system.Neuromotor interfaces use the patient’s neurological signals for control, such as brain–computer interfaces (BCI) that enable robotic control through brain activity, facilitating rehabilitation for individuals with very limited movements, or EMG (electromyographic) sensors that detect the user’s movement intentions.AI adaptation-powered interfaces automatically adapt to the user’s needs without direct intervention, using artificial intelligence algorithms to optimize therapy. These interfaces adjust the level of assistance or resistance according to the user’s progress, such as rehabilitation systems that use machine learning to customize therapeutic exercises.Location of use:
Robots used in hospital clinics or rehabilitation centers are designed to provide intensive care under the direct supervision of specialists. As an example, the BRAVO exoskeleton for upper limb rehabilitation is used in clinical environments.Some robotized systems have been adapted for use at home, providing patients the opportunity to continue therapy in the comfort of their own residence. For example, robotic gloves like HandSOME are used to improve hand dexterity.Robotic systems for parameterized and remotely controlled therapy by specialists through IoT technologies can be used in a teletherapy context.Mechanical structure:
End-effector structured devices focus on interaction with one or more specific parts of the body (end effectors) such as the hands, legs, or head. They are used in specific rehabilitation exercises, providing assisted movement and sensory feedback. The Bi-Manu-Track is an example of an end-effector device.Exoskeletons are external robotic devices that are mounted on the body, providing motor support and assistance to people with disabilities. They are especially used for rehabilitating patients with paralysis or muscle weakness, facilitating movement of the upper or lower limbs through mechanical support and sometimes electrical stimulation. Examples of exoskeletons include ARMin III and T-WREX.Wearable robotics are lightweight and flexible devices that can be worn on the body, providing continuous support in daily activities or rehabilitation exercises. They are ideal for long-term use, especially in home environments.Soft robotics use flexible and adaptable materials that mimic the natural movements of the human body. This category is designed to provide safer and more comfortable interaction with the user, making it ideal for patients requiring gentle therapy and support in precision movements.Type of actuators:
Electric motors (AC/DC) are the most common in robotic systems due to their precise control over speed and position. They are ideal for rehabilitation exercises that require fine adjustments of movement.Hydraulic and pneumatic motors provide significant force and are often used in exoskeletons or other devices that require support for body weight. However, they are less precise than electric motors and can be more challenging to control.Functional electrical stimulation (FES) involves applying electrical impulses to muscles to induce muscle contractions. It is used in combination with robotic devices to improve muscle strength and facilitate motor retraining.Control structure:
Open-loop controls are used in the initial stages of rehabilitation, where precise control is less critical. These systems operate without direct feedback, applying control actions based on a predefined set of instructions without adjusting them in real time.Closed-loop PID controls (proportional–integral–derivative) are widely used due to their simplicity and ability to provide stable and efficient control for a variety of tasks.Robust control applied in robotic rehabilitation systems utilizes a fractional approach to control a seven degrees of freedom (DoF) exoskeleton, providing efficient management of friction dynamics and disturbances. The main advantage is its advanced ability to withstand uncertainties, parameter changes, and perturbations, such as a patient’s hand tremors.Adaptive control with active disturbance rejection (ADRC) modifies its behavior to adapt to changes in system parameters or uncertain parameters. It is preferred for its ability to simplify the control system while providing advanced disturbance and uncertainty rejection capabilities.Hybrid control combines elements of open and closed loop systems, providing flexibility in treatment by adapting to different stages of rehabilitation.The type of control structure is not the primary focus of this work, as all identified types have proven to be effective.Control inputs:
Transducers for the forces and torques applied by the patient or the robot provide feedback for adjusting assistance. The major advantage arises in the measurement or generation of torque moments, which can be very precisely achieved in robotic systems compared to traditional manual therapy, where this aspect is subjective.Optical encoders are used to measure position and angular speed, ensuring precise motion control.EMG (electromyographic) signals capture muscle activity to initiate or guide robotic movement, facilitating a more natural and intuitive interaction with the user.Pressure measurements help determine the applied force and appropriately adjust the assistance in the case of hydraulic or pneumatic actuators.

### 3.2. Analysis of the Advantages and Limitations of Each Category of Robotic Systems

Advantages: Precision and repeatability of movements, continuous monitoring of progress, adaptability to patient needs, potential for improving patient motivation.

Limitations: High acquisition and maintenance costs, operational complexity, the need for specific training for medical staff, limited accessibility in some regions.

Examples:Exoskeleton: Advantages, mobility support and intensive rehabilitation; Limitations, high cost and high weight.Robotic arms: Advantages, precision in fine movements; Limitations, the need for controlled and stable space for use.

### 3.3. Investigation of Technical, Economic, Social, and Cultural Barriers

Technical: Complexity and reliability of robotic systems, integration with other medical technologies.

Economic: Initial and maintenance costs, lack of adequate funding.

Social: Acceptance by patients and medical staff, resistance to change.

Cultural: Variations in perception of technology in different cultures, language and educational barriers.

Examples:Technical: Need for constant calibration and maintenance.Economic: Limited budgets of medical institutions.Social: Negative perception of advanced technology in traditional communities.

### 3.4. Identifying Facilitative Factors for the Adoption of Robotic Technologies

Funding: Public and private funding initiatives, grants, and support programs.

Training programs: Courses and workshops for medical staff to learn how to use and maintain robotic systems.

Awareness campaigns: Informing the public and medical communities about the benefits and uses of robotic technology.

Examples:Funding: Government grants for the acquisition of robotic equipment.Training programs: Specialized courses for therapists and biomedical engineers.Awareness campaigns: Educational programs in the media and medical conferences.

## 4. Results

To understand the impact of robotic technologies on motor recovery and to compare the effectiveness of different robotic rehabilitation systems, the Fugl-Meyer Assessment (FMA) is most used. It is an essential tool in measuring post-stroke motor progress, assessing motor function, sensory function, balance, joint range of motion, and joint pain. The FMA allows a detailed assessment of motor recovery, with an upper limb scale rated at up to 66 points. The motor recovery assessment provides an objective and standardized method to measure the effectiveness of various robotic rehabilitation systems [53].

Significant Results from Selected Studies:

MIT-MANUS was one of the first systems evaluated, showing improvements in upper limb motor function without significant adverse events over approximately 500 h of operation. It was studied in a group of 96 post-stroke patients, demonstrating improvements in upper limb mobility, with positive effects observed six months after the intervention.

The study revealed that patients using the MIT-MANUS system showed substantial improvements in upper limb motor function. These improvements were quantified using standard assessment tools like the Fugl-Meyer Assessment (FMA), which measures motor recovery post-stroke [54].

Throughout the study, no significant adverse events were reported, highlighting the safety and tolerability of the MIT-MANUS system. This finding is critical, as it demonstrates that intensive robotic therapy can be conducted safely over extended periods [55].

The RATULS (Robot-Assisted Training for the Upper Limb after Stroke) trial is one of the largest and most significant research studies dedicated to evaluating the effectiveness of robotic therapy for upper limb rehabilitation after stroke. This randomized, controlled trial included 770 participants divided into three groups: robotic therapy, intensive exercise therapy, and standard care. The main objective of the study was to determine whether robotic therapy can improve upper limb functionality more effectively than traditional rehabilitation methods. It also aimed to assess the impact of robotic therapy on patients’ quality of life, personal autonomy, and degree of independence in carrying out daily activities. One group received upper limb-specific robotic therapy, the second group received intensive exercise therapy but without robot assistance, and the third group received the standard care recommended for stroke recovery. Therapy was applied over a fixed period with regular sessions, and patient progress was monitored using standardized measurements, including the Fugl-Meyer Upper Extremity Assessment (FMA-UE) score. No significant differences were observed between the robotically treated group and the conventionally treated group in terms of improvement in FMA-UE scores, suggesting that robotic therapy, as applied in the study, was not superior to standard therapy in improving upper limb function. All groups showed improvements in upper limb function, indicating that intensive interventions, whether robotic or traditional, may contribute to post-stroke recovery. Although the RATULS study did not demonstrate a clear superiority of robotic therapy over traditional rehabilitation methods, the results highlighted the importance of intensive and personalized therapy for post-stroke patients [56]. 

The ARMin exoskeleton was evaluated in a study involving 77 stroke patients, where a significant improvement in Fugl-Meyer Assessment (FMA) scores for arm and hand function was observed. ARMin III was tested in a 24-session parallel randomized trial. Results showed that the FMA-UE (Fugl-Meyer Assessment-Upper Extremity) scores of the robotically treated group were 2.6, 3.4, 3.4, and 3.1 at 4, 8, 16, and 34 weeks. Comparatively, the control group results were 2.0, 2.6, 2.8, and 2.9 at the same time intervals, indicates a significantly greater improvement for the robotically treated group, although the difference between the two groups decreased over time [57].

The ReoGo device was used in a study of 44 patients, demonstrating improvements in movement coordination and hand dexterity as measured by the Box and Block hand function test and the Nine Hole Peg test. Designed for upper limb rehabilitation, ReoGo enables repetitive and precise movements with adjustable assistance and resistance. Its interactive interface provides real-time feedback and adapts task difficulty based on performance, enhancing motor learning and neuroplasticity. ReoGo has been associated with increased patient motivation and adherence, leading to better motor recovery outcomes [58].

The Bi-Manu-Track was tested on 44 patients, with results indicating an improvement in movement symmetry between the upper limbs. Patients were divided into a control group and a group treated with an Arm Trainer (AT) robot for 30 sessions, either undergoing conventional therapy (control group) or robotic therapy using Bi-Manu-Track (AT group). In the robotic AT group, the Fugl-Meyer (FM) score was 15 points higher at the end of the study and 13 points higher at the 3-month follow-up compared to the control group [59].

T-WREX was also tested in a randomized controlled trial with an intervention conducted over 24 sessions. The robotically treated group saw an improvement in the FM score of 3.3 ± 2.4 (*p* = 0.001) and 3.6 ± 3.9 after a 6-month follow-up (*p* = 0.005), compared to the control group’s FM improvement of 2.2 ± 2.6 (*p* = 0.004) and 1.5 ± 2.7 after a 6-month follow-up (*p* = 0.06) [60].

In a secondary analysis of data from the multicenter VA-ROBOTICS study, the effectiveness of intensive therapy, including both robot-assisted therapy and intensive conventional therapy, was examined compared to standard care in improving motor function in chronic stroke patients. The research included 127 patients with moderate to severe upper limb deficits, at least six months post-stroke. They were randomly assigned to three groups: a robot-assisted therapy group (49 participants), a comparative intensive care group (50 participants), and a standard care group (28 participants) [27].

Robot-assisted therapy was administered using InMotion rehabilitation robots, which allow guided movements of the upper limb, providing visual and kinesthetic feedback to patients. Intensive comparative therapy was adapted to match robotic therapy in terms of duration and frequency of sessions, but was based on traditional rehabilitation methods. The standard care group received treatments commonly available in Veterans Affairs medical centers, without a specific intervention dictated by the study. Study results indicated a significant benefit of intensive therapy over standard care at 12 weeks post-treatment, with a mean improvement of 4.0 points in the Fugl-Meyer score, which assesses sensorimotor recovery (95% CI = 1.3–6.7; *p* = 0.005). However, at 36 weeks post-treatment, the observed benefit was attenuated, with a mean difference of 3.4 points (95% CI = −0.02 to 6.9; *p* = 0.05). Subgroup analyses revealed that younger patients and those whose time since stroke was shorter showed greater improvements both immediately and in the long term, highlighting the importance of personalized treatment based on each patient’s individual characteristics [61].

Since stroke is the leading cause of motor disability, most studies in the area have analyzed the efficacy of post-stroke robot-assisted therapy. In a meticulous evaluation of data from 44 randomized controlled trials involving a total of 1362 participants, modest but significant improvements in motor function and muscle strength of the affected arm were observed with no serious adverse events reported, indicating the safety of this therapeutic modality [62]. The studies reviewed offered a variety of interventions, including the use of specialized robotic systems for shoulder and elbow rehabilitation, as well as elbow and wrist devices, reflecting the technological diversity in the field of robotic rehabilitation [63]. Improvements in motor control, as assessed by mean changes of approximately 2 points on the Fugl-Meyer arm scale, and increased muscle strength highlighted the specific effectiveness of the targeted joint interventions [64]. However, these effects did not generalize to global upper limb capacity or to the performance of basic daily activities, highlighting the need for more holistic and personalized therapeutic approaches. Despite the observed improvements, the analysis revealed that the effects of robotic therapy in the initial weeks after a stroke remain unclear, pointing to a gap in the literature and in the understanding of the optimal time to initiate this form of therapy. Additionally, the interactions between patient age, time since stroke, and type of robotic intervention suggest that the effectiveness of therapy may vary depending on individual patient characteristics, highlighting the importance of careful patient selection and personalization of therapy.

In recent research on gait rehabilitation after stroke, Chung BP’s 2017 study included a total of 41 participants, 14 of whom were assigned to the group receiving robotic-assisted gait therapy (RAGT) and 27 to the control group, which received standard physical therapy. Results showed that the group receiving robot-assisted therapy benefited to a greater extent in terms of ambulation, motor activity, and balance, suggesting additional benefits of robot-assisted therapy [65].

In 2018, ref. [66] explored the impact of combining robot-assisted training with conventional physical therapy. The study concluded that the combined management provided more substantial benefits for patients with chronic hemiplegia, indicating the effectiveness of adding robotic therapy to rehabilitation plans.

In 2021, ref. [67] compared the effects of using exoskeletal robots for gait training with traditional therapy in a study involving 130 participants split between a group that used robotic technology for gait training and a control group that received traditional gait training. No significant differences were found between the groups in terms of improved locomotion skills, suggesting similar effectiveness between robotic and traditional therapy.

One study by Pohl et al. included 155 patients who were divided into two groups: one received locomotor training and physical therapy, and the other only physical therapy. The group that benefited from both interventions showed significant improvements in walking ability, highlighting the importance of intensive and repetitive training.

A meta-analysis conducted in 2020 by [68], which included 62 trials with a total of 2440 individuals, evaluated the effects of electromechanically assisted gait training in combination with physical therapy. The findings indicated that this type of training, in combination with physical therapy, increased the likelihood of unassisted walking compared to traditional methods, illustrating the potential benefits of robotic technology in motor rehabilitation.

A notable example of exoskeleton technology in the rehabilitation of physical disabilities is the REX exoskeleton, which has been evaluated for its effectiveness in improving balance and lower limb function in sub-acute stroke patients. In a pilot randomized controlled trial conducted by Yuting Zhang et al. (2024), the use of the REX exoskeleton led to significant improvements in both static and dynamic balance, as well as in lower limb motor function, when compared to a control group receiving standard physical therapy without robotic assistance [69]. The study’s methodology involved several weeks of rehabilitation sessions where patients in the experimental group utilized the REX exoskeleton. The results were measured using established clinical scales such as the Berg Balance Scale (BBS) and the Timed Up and Go (TUG) test. Patients in the REX group showed a marked improvement, with an average increase of 5 points on the BBS and a reduction of 4 s in TUG test times, compared to 2 points and 1 s improvements in the control group. These findings underscore the potential of exoskeletons like REX to facilitate motor recovery by providing repetitive, controlled movements that are crucial for promoting neuroplasticity and motor relearning. The implications of these results are significant, suggesting that early integration of exoskeleton-assisted rehabilitation could enhance the recovery trajectory for stroke patients. The precision and repeatability offered by the REX exoskeleton allow for targeted interventions that address specific motor deficits, making it a valuable addition to conventional rehabilitation methods [69].

## 5. Discussion and Conclusions

### 5.1. Discussion

The analyzed data highlights the potential of robotic systems in improving the functional recovery of patients with motor disabilities, highlighting the significant progress made in the field of robotic rehabilitation. The mentioned clinical studies show measurable improvements in upper limb motor function, validating the effectiveness of robot-assisted therapy in the rehabilitation context. 

These results can be used to argue the importance of further research and development in the field of robotic rehabilitation.

#### 5.1.1. Full Brain–Computer Interface (BCI) Integration

The integration of brain–computer interface (BCI) technology with exoskeletons represents a significant advancement in the field of motor rehabilitation, offering the potential to restore motor functions in patients with severe physical disabilities. A recent study by Kueper et al. (2024) explored the application of transfer learning on EEG data to improve motor intention recognition without the need for specific calibration sessions [70]. This approach allows for real-time control of exoskeletons purely through brain activity, offering a more intuitive and seamless interaction between the user and the device. The study demonstrated that integrating BCI with exoskeletons significantly enhances both the accuracy and the comfort of the rehabilitation process. Patients were able to achieve more precise control over their movements, which is crucial for effective motor recovery. The ability to bypass lengthy and cumbersome calibration sessions is particularly beneficial in clinical settings, where time efficiency and patient adaptability are critical factors. This integration of BCI technology not only improves the effectiveness of exoskeleton-assisted rehabilitation but also broadens its applicability to a wider range of patients, including those with severe motor impairments who might otherwise be unable to benefit from traditional rehabilitation methods. As such, the development of BCI technologies represents a promising frontier in enhancing the capabilities and accessibility of robotic rehabilitation systems [70].

Advances in the development of brain–computer interfaces for controlling robotic devices have shown promise, yet their full and effective implementation in motor rehabilitation is still limited. Studies indicate that while BCIs can potentially facilitate motor recovery by directly translating brain signals into robotic movements, challenges related to signal accuracy, user comfort, and integration with existing rehabilitation protocols remain [71]. Further research to improve the accuracy and reliability of BCIs should be performed, along with the development of more effective non-invasive methods and their integration into complex robotic systems for rehabilitation. This includes advancing machine learning algorithms to better interpret brain signals and creating more user-friendly BCI devices that patients can use comfortably [72].

#### 5.1.2. Personalizing Robotic Therapies

Most current robotic rehabilitation systems provide standardized treatment protocols, which may not be effective for all patients due to individual differences [8].

Advanced adaptive algorithms should be developed that can adjust therapy in real time based on patient progress and responses, using artificial intelligence and machine learning to enhance treatment efficacy [73].

#### 5.1.3. Accessibility and Costs of Robotic Systems

While robotic technologies, including exoskeletons and brain–computer interface (BCI) systems, have shown significant promise in enhancing motor rehabilitation outcomes, there are several challenges that impede their broader implementation. Among these, the high production cost of such advanced systems stands out as a significant barrier. The complexity of design, engineering, and regulatory compliance drives up costs, making these technologies less accessible to a wide range of healthcare providers and patients. The development and deployment of robotic systems requires substantial financial investment, particularly in the production of high-precision components such as sensors and actuators, as well as in software development and validation processes. For research institutions and smaller companies, these costs can be prohibitive, often restricting innovation and limiting the advancement of new technologies beyond the prototype stage. Moreover, the cost barrier extends to the healthcare facilities that may hesitate to adopt expensive robotic systems without definitive proof of cost-effectiveness. This financial constraint can slow the diffusion of these technologies, potentially leaving many patients without access to the most advanced rehabilitation options available.

Addressing these cost-related challenges requires a concerted effort to develop more affordable production methods, explore economies of scale, and foster partnerships that can help distribute the financial burden more evenly across the industry. Research should be conducted to reduce production costs and develop sustainable economic models, and initiatives should be taken to implement subsidy programs and secure funding from government and non-governmental organizations in order to enhance the accessibility of robotic equipment in healthcare institutions.

#### 5.1.4. Integration into Clinical Practice

The effective integration of robotic technologies into the daily routines of clinics and hospitals is often hampered by a lack of adequate infrastructure and staff training [74]. 

In the implementation of robotic systems in clinical practice, one of the most significant concerns is the potential for malfunctions due to hardware failures, software bugs, environmental factors, or human errors. These malfunctions can have serious implications for patient safety and the effectiveness of the rehabilitation process. For instance, hardware failures such as motor or sensor malfunctions could lead to improper or unsafe movement patterns, risking further injury to the patient. Software bugs might cause delays or incorrect responses from the robotic system, undermining therapeutic goals. Environmental factors, such as power surges or temperature fluctuations, could also disrupt the functioning of these sophisticated devices, while human errors, including improper setup or handling, might lead to system failures or suboptimal performance.

Thus, it is important to develop comprehensive training programs for therapists and medical staff, invest in clinical infrastructure, and promote partnerships between technology developers and medical institutions [75].

It is essential to incorporate robust fail-safe mechanisms, regular maintenance protocols, and thorough staff training. Additionally, designing systems with built-in redundancy can help ensure that even in the event of a failure, the robot can continue to operate safely or shut down in a controlled manner, minimizing the risk to the patient. These advancements highlight the versatility of robotic technologies and their potential to address various challenges in healthcare, from improving surgical outcomes to supporting aging populations in maintaining independence. Future research should continue to explore and expand these applications, ensuring that robotic systems are integrated seamlessly into different aspects of healthcare delivery.

#### 5.1.5. Long-Term Evaluation of the Effectiveness of Robotic Therapies

Existing studies often focus on short-term outcomes without evaluating the long-term effectiveness of robotic-assisted rehabilitation.

In future, long-term clinical trials that monitor and evaluate the impact of robotic therapies on patients’ functional recovery as well as their quality of life should be implemented [74].

#### 5.1.6. Social and Cultural Acceptance of Robotic Technologies

Cultural and social resistance to the use of robotic technology can limit the widespread adoption of these solutions.

This necessitates the delivery of awareness and education campaigns to demystify robotic technologies and highlight their benefits, and the involvement of communities in the development and implementation process of these technologies.

### 5.2. Conclusions

This review of advances and challenges in the field of robotic motor rehabilitation brings to the fore the huge potential of emerging technologies in improving the lives of patients with physical disabilities. Results from selected studies underscore the effectiveness of robotic systems in improving motor function of the upper and lower limbs, offering new hope for recovery. However, we find that in spite of these measurable improvements in motor control and muscle strength, the direct applicability of these improvements in patients’ daily activities and autonomy remains a challenge.

Studies, such as those mentioned in the article, illustrate significant technological advances, but also the critical need for additional research to address gaps in the literature.

Although robotic systems show significant benefits in improving motor function, further research is essential to maximize the effectiveness of these technologies in clinical practice. It is imperative that greater attention is paid to personalizing therapy, considering the individual specificities of each patient. It is also vital to address the challenges of cost, staff training, and adapting infrastructures to facilitate the widespread integration of robotic solutions in motor rehabilitation.

In addition to the functional improvements observed because of robotic therapy, it is essential to recognize and harness the potential of these technologies to improve the psychological state of patients. Tailored approaches that consider the individual needs of patients, along with innovative solutions to overcome cost and accessibility barriers, are key to the wider integration of robotic rehabilitation into clinical practice. By expanding research and development in this area, we can profoundly transform the motor recovery and quality of life of patients with physical disabilities.

Even if the increase in scores assessing the effectiveness of robotic therapy does not yet represent a major difference from classical therapy, some advantages remain unbeatable: the reduction of staff required, the possibility of telemedicine and continuous treatments without the need to travel or staff changes, and of course the precision with which robotic systems can measure and generate, in particular, the specific torques and velocities of certain movements in physical rehabilitation exercises.

## Figures and Tables

**Figure 1 healthcare-12-01720-f001:**
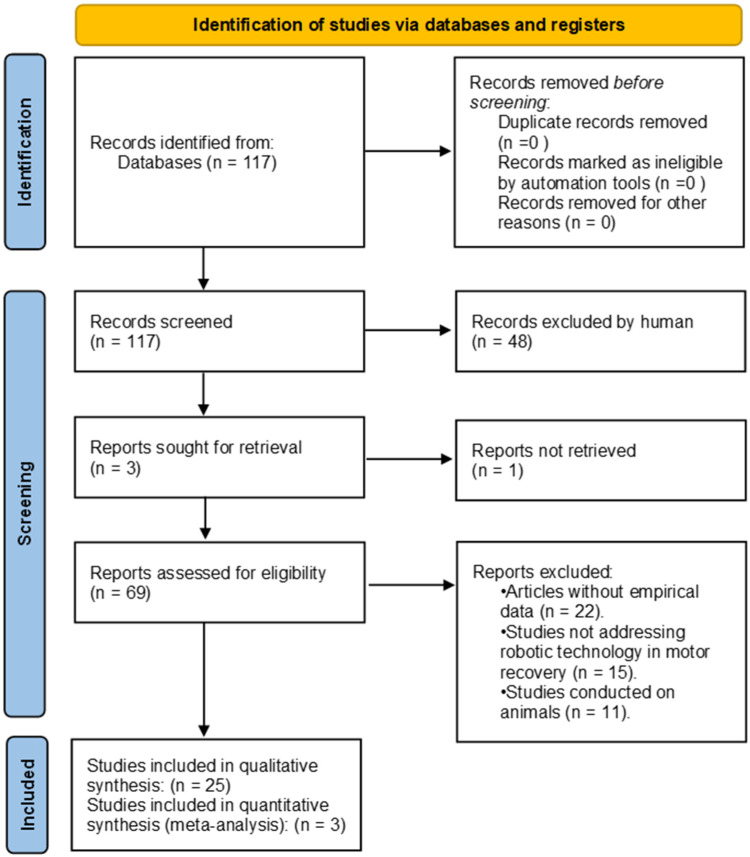
PRISMA flow diagram.

**Table 1 healthcare-12-01720-t001:** Selected papers for the study.

Type	Description	Reference
Systematic reviews and meta-analyses	Synthesizes the results from multiple primary research studies on a specific topic, providing a comprehensive assessment of the availability and quality of evidence. Examines the overall effectiveness of robot-assisted therapy in upper limb or gait rehabilitation.	[6,7,8,9,10,11,12,13,14,15,16,17,18,19,20]
Randomized Controlled Trials (RCTs), Case and Pilot Studies	Compares the effectiveness of an intervention with a control group in a well-structured setting with randomly selected participants. Directly tests the effects of robot-assisted interventions compared to standard rehabilitation methods. Explores the impact of interventions on individual cases or small groups, providing initial data for further research. Initial use of robotic therapies for specific patients or in preliminary settings.	[21,22,23,24,25,26,27]
Exploratory Studies and Technical Development Analyses	Focuses on the development, improvement, and evaluation of technical aspects of robotic devices, including design and implementation. Describes technological innovations in robotic rehabilitation or analyzes specific aspects of system performance. Investigates the potential and practicality of novel approaches in a research setting, often in the early stages. Tests new methodologies or technologies in rehabilitation to determine the viability of more rigorous future studies.	[28,29,30,31,32,33,34,35,36,37,38,39,40,41,42,43,44,45]
Robotic systems for patient care	Investigates robotic systems used for dressing, eating, or washing.	[46,47,48,49]

## Data Availability

Not applicable.

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
