# Peer review of "Robotics in Physical Rehabilitation: Systematic Review"

_healthcare, 2024, doi:10.3390/healthcare12171720_

Round 1

Reviewer 1 Report (Previous Reviewer 2)

Comments and Suggestions for Authors

The revised version is reasonably okay.

Author Response

Thank you for your feedback. We appreciate your positive assessment of the revised version. In response to your comments, we have made additional improvements to further enhance the quality of the manuscript.

Reviewer 2 Report (Previous Reviewer 3)

Comments and Suggestions for Authors

the revised version and the author’s reply to comments are appropriate, and the paper can be accepted in its current form.

Comments on the Quality of English Language

.

Author Response

Thank you for your review and positive evaluation. We are glad that the revisions and responses met your expectations. We appreciate your recommendation for acceptance in its current form.

Reviewer 3 Report (New Reviewer)

Comments and Suggestions for Authors

The article entitled “Robotics in Physical Rehabilitation – A Systematic Review” shows a comprehensive review on recent advancements for physically disability.

The main contribution of this research work is showcasing the robotic technologies applied in motor related problems and also, shows the recent technology of Exoskeleton, BCI.

However, the article shows a detailed review, I strongly recommend to integrate the following points.

Suggestions:

1.     Can show a use-case of how exoskeleton assist physical disability.

2.     Adding a detailed note on Full Brain-computer Interface integration in 5.1.1, will showcase how the development of BCI improves the rehabilitation.

3.     Discuss how the Production cost hinders the research in robotic application.

4.     What if robots malfunction due to hardware failures, software bugs, environmental factors, or human errors, and Discuss its impact.

5.     There are numerous developments in Robotic application in healthcare domain, but I feel the article lacks in wide discussion on those applications.

6.     With a neat sketch, Illustrate the robot’s application in healthcare domain.

7.     Suggestion to add these references;

a.      Exoskeleton rehabilitation robot training for balance and lower limb function in sub-acute stroke patients: a pilot, randomized controlled trial: https://doi.org/10.1186/s12984-024-01391-0

b.     Avoidance of specific calibration sessions in motor intention recognition for exoskeleton-supported rehabilitation through transfer learning on EEG data: DOI: 10.1038/s41598-024-65910-8

Author Response

Dear Reviewer 3,

Thank you for your positive evaluation and valuable suggestions, which we have addressed in detail below.

Comment 1: Can show a use-case of how exoskeleton assist physical disability.

  1. Reply: We have added the following text, in chapter 4, lines 504-522:
    “A notable example of exoskeleton technology in the rehabilitation of physical dis-abilities is the REX exoskeleton, which has been evaluated for its effectiveness in im-proving balance and lower limb function in sub-acute stroke patients. In a pilot randomized controlled trial conducted by Yuting Zhang et al. (2024), the use of the REX exoskeleton led to significant improvements in both static and dynamic balance, as well as in lower limb motor function, when compared to a control group receiving standard physical therapy without robotic assistance. [69] The study's methodology involved several weeks of rehabilitation sessions where patients in the experimental group utilized the REX exoskeleton. The results were measured using established clinical scales such as the Berg Balance Scale (BBS) and the Timed Up and Go (TUG) test. Patients in the REX group showed a marked improvement, with an average increase of 5 points on the BBS and a reduction of 4 seconds in the TUG test times, compared to 2 points and 1 second improvements in the control group. These findings underscore the potential of exoskeletons like REX to facilitate motor recovery by providing repetitive, controlled movements that are crucial for promoting neuroplasticity and motor re-learning. The implications of these results are significant, suggesting that early integration of exoskeleton-assisted rehabilitation could enhance the recovery trajectory for stroke patients. The precision and repeatability offered by the REX exoskeleton allow for targeted interventions that address specific motor deficits, making it a valuable addition to conventional rehabilitation methods [69].”

Comment 2: Adding a detailed note on Full Brain-computer Interface integration in 5.1.1, will showcase how the development of BCI improves the rehabilitation.

  1. Reply: We have added the following text, in paragraph 5.1.1, lines 534-551:
    “The integration of Brain-Computer Interface (BCI) technology with exoskeletons represents a significant advancement in the field of motor rehabilitation, offering the potential to restore motor functions in patients with severe physical disabilities. A recent study by Kueper et al. (2024) explored the application of transfer learning on EEG data to improve motor intention recognition without the need for specific calibration sessions [70]. This approach allows for real-time control of exoskeletons purely through brain activity, offering a more intuitive and seamless interaction between the user and the device. The study demonstrated that integrating BCI with exoskeletons significantly enhances both the accuracy and the comfort of the rehabilitation process. Patients were able to achieve more precise control over their movements, which is crucial for effective motor recovery. The ability to bypass lengthy and cumbersome calibration sessions is particularly beneficial in clinical settings, where time efficiency and patient adaptability are critical factors. This integration of BCI technology not only improves the effectiveness of exoskeleton-assisted rehabilitation but also broadens its applicability to a wider range of patients, including those with severe motor impairments who might otherwise be unable to benefit from traditional rehabilitation methods. As such, the development of BCI technologies represents a promising frontier in enhancing the capabilities and accessibility of robotic rehabilitation systems [70].”

Comment 3: Discuss how the Production cost hinders the research in robotic application.

  1. Reply: We have added the following text, in paragraph 5.1.3, lines 570-588:
    “While robotic technologies, including exoskeletons and Brain-Computer Interface (BCI) systems, have shown significant promise in enhancing motor rehabilitation outcomes, there are several challenges that impede their broader implementation. Among these, the high production cost of such advanced systems stands out as a significant barrier. The complexity of design, engineering, and regulatory compliance drives up costs, making these technologies less accessible to a wide range of healthcare providers and patients. The development and deployment of robotic systems require substantial financial investment, particularly in the production of high-precision components such as sensors and actuators, as well as in software development and validation processes. For research institutions and smaller companies, these costs can be prohibitive, often restricting innovation and limiting the advancement of new technologies beyond the prototype stage. Moreover, the cost barrier extends to the healthcare facilities that may hesitate to adopt expensive robotic systems without definitive proof of cost-effectiveness. This financial constraint can slow the diffusion of these technologies, potentially leaving many patients without access to the most advanced rehabilitation options available.

What should be done: Addressing these cost-related challenges requires a concerted effort to develop more affordable production methods, explore economies of scale, and foster partnerships that can help distribute the financial burden more evenly across the industry.”

Comment 4: What if robots malfunction due to hardware failures, software bugs, environmental factors, or human errors, and Discuss its impact.

  1. Reply: We have added the following text, in paragraph 5.1.4, lines 595-604 and 609-617: “In the implementation of robotic systems in clinical practice, one of the significant concerns is the potential for malfunctions due to hardware failures, software bugs, environmental factors, or human errors. These malfunctions can have serious implications for patient safety and the effectiveness of the rehabilitation process. For instance, hardware failures such as motor or sensor malfunctions could lead to improper or unsafe movement patterns, risking further injury to the patient. Software bugs might cause delays or incorrect responses from the robotic system, undermining therapeutic goals. Environmental factors, such as power surges or temperature fluctuations, could also disrupt the functioning of these sophisticated devices, while human errors, including improper setup or handling, might lead to system failures or suboptimal performance.”

and

“It is essential to incorporate robust fail-safe mechanisms, regular maintenance protocols, and thorough staff training. Additionally, designing systems with built-in redundancy can help ensure that even in the event of a failure, the robot can continue to operate safely or shut down in a controlled manner, minimizing the risk to the patient. These advancements highlight the versatility of robotic technologies and their potential to address various challenges in healthcare, from improving surgical outcomes to supporting aging populations in maintaining independence. Future research should continue to explore and expand these applications, ensuring that robotic systems are integrated seamlessly into different aspects of healthcare delivery.”

Comment 5: There are numerous developments in Robotic application in healthcare domain, but I feel the article lacks in wide discussion on those applications.

  1. Reply: We have added the following text, in chapter 1, lines 108-121:
    “Robotics has emerged as a transformative technology across various healthcare domains, extending far beyond its initial applications in surgical assistance. For instance, robotic systems like the Da Vinci robot have revolutionized minimally invasive surgery by providing surgeons with enhanced precision and control. In patient care, assistive robots are increasingly used to support daily activities for the elderly or individuals with disabilities, such as helping with mobility, feeding, and dressing. Additionally, in the field of telemedicine, robots have played a crucial role in remote monitoring and consultations, especially in underserved regions and during global health crises like the COVID-19 pandemic. The diversity of these systems, ranging from surgical robots to assistive and telemedicine robots, highlights their potential to address a variety of healthcare challenges. While this review will focus primarily on the applications of robotic systems in motor rehabilitation, it is essential to recognize that these technologies are part of a much larger movement towards the integration of robotics in healthcare, promising to enhance both the quality and accessibility of medical care across the globe.”

Comment 6: With a neat sketch, Illustrate the robot’s application in healthcare domain.

  1. Reply: The response given at 5, provides the reader with a clear understanding of medical applications that utilize robots. However, the use of images depicting such existing systems requires obtaining permission to use them.

Comment 7: Suggestion to add these references;

  1. Reply: We have included the two references you suggested in the article at positions 69 and 70. One for a successful example of exoskeleton usage (suggestion 1) and the other for BCI technology (suggestion 2).

[69]       Zhang, Y., Zhao, W., Wan, C. et al. Exoskeleton rehabilitation robot training for balance and lower limb function in sub-acute stroke patients: a pilot, randomized controlled trial. J NeuroEngineering Rehabil 21, 98 (2024). https://doi.org/10.1186/s12984-024-01391-0

[70]       Kueper, N., Kim, S.K. & Kirchner, E.A. Avoidance of specific calibration sessions in motor intention recognition for exoskeleton-supported rehabilitation through transfer learning on EEG data. Sci Rep 14, 16690 (2024). https://doi.org/10.1038/s41598-024-65910-8

This manuscript is a resubmission of an earlier submission. The following is a list of the peer review reports and author responses from that submission.

Round 1

Reviewer 1 Report

Comments and Suggestions for Authors

Title:

Authors reviewed literature about robotics in motor recovery/rehabilitation and physical disability. Therefore, the title should be adjusted toward the review goal. Currently the title is very broad, and the review do not cover that broad title.

Abstract:

Rewrite to be clearer, and more organized.

Introduction:

Line 31: ‘’multiple sclerosis and degenerative diseases’’

MS is increasingly recognized as a neurodegenerative disease triggered by an inflammatory attack of the CNS, therefore the above statement should be modified.

References in the introduction: All the introduction with 5 references? The references are scarce, currently several paragraphs are without references, therefore references should be adequately added.

This applied to the throughout of the review

Line 44: You mentioned robotics, and then this disappeared to line 68, where again authors mentioned about robotics and in between all statements were mostly repeated. Authors should adjust and be consistent and concise.

Objectives and methodology

What is the type of this review?

‘’The overall objective of the study is to conduct an in-depth investigation of how innovations in robotics can facilitate motor recovery and help reduce the workload of care giving staff, thereby improving the quality of life for patients with physical disabilities, establishing the following specific objectives.’’

In the abstract:

‘’ This analysis aims to evaluate the progress and challenges encountered in implementing robotic technology. ‘’

Authors should adjust the objective of their review and report it correctly.

Method is poorly written and no clear strategy of what they followed to search for their included studies. What were the databases? Quality assessment? Risk of bias?

Table 1: There is no mention of table 1 and no caption for it.

4. Location of use:

Line 212: since you are saying location, then it should be clinic, not ‘’clinical’’

Results are mixed up with the methods section!

Results section is not clear and need to be more organized and concise

Again, results section is mixed up with the discussion, where no discussing section is missing.

Comments on the Quality of English Language

Minor edits

Reviewer 2 Report

Comments and Suggestions for Authors

In this work, Banyai & Brisan present a comprehensive review on the essential roles of robotics in rehabilitation. The advancement of robotics technology has made this possible, but the exact efficacy and the mass-adoption in clinical practice are yet to be seen. I appreciate the diversity in the classification of robots in rehabilitation outlined by the authors, as well as barriers and facilitative factors which are real and practical. There are parts to be revised to ensure the quality of the content, particularly Section 4 (Results). Please consider the following feedback.

1. Abstract: Please rephrase Line 8-11 as the sentence is too long. Please mention how many articles were synthesized for your review.

2. Introduction, Line 75: Can the authors introduce why those sets of feedback are important? What about the role of positive or reward feedback? It is an essential part of motor (re)learning process and it has been brought over to rehab. I invite the authors to the following relevant paper

3. Perhaps the authors shall be more focused in the introduction. Is it possible to make the Introduction section more condensed? E.g. why did the authors discuss about psychological aspects if the robots are not solving that (Line 57)?

4. Line 138-147: Not readable for between inclusion/exclusion, either add a space before exclusion section, or make it as a table instead.

5. Part (3): Is this the section where you define the scope of what constitutes robotics? If yes, can the authors state this at the opening sentence? Often, clinicians and researchers are unsure about the definition. For example: is treadmill system a robot?

6. Clearing the scope is also good. For example, what did the authors mean by physical, it is not merely motor, but also elements of sensation (e.g. touch) and the ability to exert force/dynamic aspects.

7. Part (4) Results: While the presentation of the different RCTs (e.g.  RATULS) is interesting, the authors shall be more careful not to make the content to be of personal opinions. For example: Sub-section 4.1 to 4.6 are too brief. How did the authors come up with the challenges and what should be done (recommendation)? In each recommendation, I suggest the authors provide 1-2 relevant references.

Reviewer 3 Report

Comments and Suggestions for Authors

Does the review effectively compare the effectiveness of robotic therapies to traditional methods?

How does the review address the potential for bias in favor of newer technologies?

  • Are the cost, accessibility, and personalization challenges of robotic rehabilitation adequately explored?
  • Does the review discuss potential solutions or ongoing research addressing these challenges?
  • Are there ethical considerations or safety concerns regarding robotic therapy that are mentioned?
  •  
  • Does the review provide a balanced perspective on the potential and limitations of robotics in rehabilitation?
  • Is the target audience (e.g., researchers, clinicians, general public) considered in the writing style and detail provided?
  • Does the review offer clear and concise conclusions about the current state and future of robotic rehabilitation?
Comments on the Quality of English Language

After corrections only.